# Dietary Polyphenols, Microbiome, and Multiple Sclerosis: From Molecular Anti-Inflammatory and Neuroprotective Mechanisms to Clinical Evidence

**DOI:** 10.3390/ijms24087247

**Published:** 2023-04-14

**Authors:** Giuliana La Rosa, Maria Serena Lonardo, Nunzia Cacciapuoti, Espedita Muscariello, Bruna Guida, Raffaella Faraonio, Mariarosaria Santillo, Simona Damiano

**Affiliations:** 1Dipartimento di Medicina Clinica e Chirurgia, Università di Napoli “Federico II”, 80131 Naples, Italy; 2Dipartimento di Medicina Molecolare e Biotecnologie Mediche, Università di Napoli “Federico II”, 80131 Naples, Italy

**Keywords:** polyphenols, multiple sclerosis, neuroprotection, microbiome, resveratrol, curcumin, luteolin, quercetin, hydroxytyrosol, epigallocatechin gallate

## Abstract

Multiple sclerosis (MS) is a multifactorial, immune-mediated disease caused by complex gene-environment interactions. Dietary factors modulating the inflammatory status through the control of the metabolic and inflammatory pathways and the composition of commensal gut microbiota, are among the main environmental factors involved in the pathogenesis of MS. There is no etiological therapy for MS and the drugs currently used, often accompanied by major side effects, are represented by immunomodulatory substances capable of modifying the course of the disease. For this reason, nowadays, more attention is paid to alternative therapies with natural substances with anti-inflammatory and antioxidant effects, as adjuvants of classical therapies. Among natural substances with beneficial effects on human health, polyphenols are assuming an increasing interest due to their powerful antioxidant, anti-inflammatory, and neuroprotective effects. Beneficial properties of polyphenols on the CNS are achieved through direct effects depending on their ability to cross the blood-brain barrier and indirect effects exerted in part via interaction with the microbiota. The aim of this review is to examine the literature about the molecular mechanism underlying the protective effects of polyphenols in MS achieved by experiments conducted in vitro and in animal models of the disease. Significant data have been accumulated for resveratrol, curcumin, luteolin, quercetin, and hydroxytyrosol, and therefore we will focus on the results obtained with these polyphenols. Clinical evidence for the use of polyphenols as adjuvant therapy in MS is restricted to a smaller number of substances, mainly curcumin and epigallocatechin gallate. In the last part of the review, a clinical trial studying the effects of these polyphenols in MS patients will also be revised.

## 1. Introduction

### 1.1. Multiple Sclerosis

Multiple sclerosis (MS) is a chronic inflammatory demyelinating disease where the dysfunction of the immune system plays an essential role. 

In 2020, a global MS prevalence of 2.8 million people was estimated, a 30 percent increase from 2013, with the highest rate of increase in prevalence recorded in the United States of America. Taking global data into account, women are twice as likely to have MS as men. However, in some countries, the ratio of women to men is 4:1 [1]. 

Recent studies have shown a significant positive association between latitude and MS prevalence. In particular, the age of disease onset is earlier in individuals living in higher latitudes (50.0–56.0°), such as regions of European origin, than in those living in lower latitudes (19.0–39.9°) [2]. This association may also be due to nutritional factors including variations in vitamin D levels [3]. Nowadays, the clinical classification of MS into relapsing and progressive forms, each of which can progress with or without activity (assessed both by MRI and clinically), is the one that seems to be most reflective of the clinical and pathophysiological reality of MS [4]. 

The characteristic hallmarks are demyelination, axonal damage [5], gliosis, and neuronal loss [6], consequences of two main pathophysiological processes: focal inflammation resulting in macroscopic inflammatory plaque (areas of focal demyelination present in the white matter of the brain and spinal cord) formation and blood-brain barrier (BBB) injury and neurodegeneration [7]. 

Depending on the location and extent of the lesions, there may be a wide parade of symptoms. The main ones include visual symptoms related to optic neuritis such as diplopia; vestibular symptoms such as vertigo; bulbar dysfunction symptoms such as dysarthria or dysphagia; motor sensory symptoms; cognitive symptoms such as memory impairment or impaired executive functions; or even psychiatric symptoms such as depression and anxiety [7].

The altered permeability of the BBB results in the migration of immune cells (CD4+ and CD8+ T lymphocytes) into the central CNS, leading to the formation of demyelinating plaques. Lymphocytes produce many molecules involved in the development and regulation of immune responses, including cytokines. Th1 lymphocytes produce interferon γ (IFN-γ) which promotes macrophage activation and the production of reactive oxygen species (ROS) and reactive nitrogen species (RNS) that mediate damage to surrounding tissues; in addition, they produce interleukin 12 (IL-12), which in turn causes an increase in INF γ and stimulates TNF-α production. Th17 lymphocytes, on the other hand, produce interleukin 17(IL-17), 21 (IL-21), and 22 (IL-22), which are also involved in the development of inflammation [8]. The development of neurodegeneration in MS is a process related to the production of proteolytic enzymes and the activation of pro-apoptotic pathways and an alteration of mitochondrial function resulting in an increased proportion of ROS, which is damaging to both neurons and glia [9,10,11]. Furthermore, the inflammatory component in MS causes not only axonal and neuronal loss, but also the activation of the degenerative cascade typical of the early phase of MS [12]. 

These pieces of evidence indicate that reducing oxidative stress and the inflammatory state should be the main goal of therapies able to improve the course of the disease [13,14,15]. The cornerstone of MS treatment is disease-modifying drugs. The first to be used, starting in the 1990s, were preparations of interferon beta and glatiramer acetate [4,16]. Over the years, as knowledge has advanced regarding the pathophysiological mechanisms underlying the disease, new molecules have also been investigated for therapeutic purposes. Indeed, in addition to the crucial role of T cells, an important step forward has been the development of disease models that more closely mimic the tissue damage pattern of MS, leading to a new awareness of the importance of humoral immunity in the pathogenesis of MS [17], which can be used as therapeutic targets.

Although a detailed description of the drugs currently used for the treatment of MS is beyond the scope of this review, it is worth providing an overview.

Despite significant advances in the treatment of MS and the fact that current disease-modifying therapies (DMT) are relatively effective in reducing new lesions and clinical relapse, the agents that were supposed to halt the underlying neurodegeneration [18] have been less effective and, most importantly, rates of progressive disability and early mortality are still alarmingly high [19]. 

Currently, in addition to injectable and orally administered drugs targeting the most common pathophysiological mechanisms underlying MS, studies in animal models involving the use of bone marrow transplantation or the administration of molecules with remyelinating activity are ongoing. Moreover, although the therapeutic present and future belong to monoclonal antibodies due to their apparently high efficacy, their potential side effects on CNS immune surveillance and the host immune system are not yet well known. Therefore, it is imperative to closely monitor patients administered these drugs to intercept early and accurately highlight potential damages [19]. 

Side effects associated with currently used therapies are divided into ‘desirable’, which require monitoring as they involve immunomodulation/immunosuppression, and ‘undesirable’, which cause hepatotoxicity, cardiotoxicity, and allergic reactions [4].

As previously mentioned, the inflammation of the CNS is the primary cause of damage in MS, even if the specific factors triggering inflammation are still unknown. This disease can also be triggered by environmental factors in individuals with complex genetic risk profiles. In addition to genetic factors, in recent years new and interesting mechanisms, such as epigenetic modifications of neuronal and glial DNA, have been linked to MS neurodegeneration [20]. Epigenetic mechanisms, such as hypomethylation, lead to the addition or removal of different chemical groups that, depending on the case, activate or inhibit gene expression and block, for example, the production of inflammatory proteins. In this sense, dietary intervention can play a key role in modulating several transcription factors (TFs) and gene expression [21]. 

Dietary factors may exert beneficial effects on inflammation, neuroprotection, and repair in MS, through several possible pathways. Food metabolites, both derived directly from the diet or produced by gut bacteria after food ingestion, exert effects through shared mechanisms. Furthermore, the diet is made of many components ingested together, so there are likely significant interactions. Peculiar dietary patterns may influence the activity and progression of MS differently [18,22]. In fact, Noormohammadi M. et al. showed that adhering to the MIND diet (Mediterranean-DASH Intervention for Neurodegenerative Delay) may be an effective strategy for MS prevention: according to research, the polyphenols found in vegetables and beans may inhibit the production of proinflammatory factors, so the use of plant-based meals and the reduction of animal-derived or high-saturated fat foods consumption is encouraged [23]. Although there are trials showing beneficial health outcomes for energy-restricted/intermittent fasting diets, the ketogenic diet, and the modified paleolithic diet, these dietary interventions may not be sustainable in the long term as they may cause deficiencies of various nutrients. Thus, the Mediterranean diet (MD) is more recommendable than other ones, because of its positive health effects supported by long-term studies and the absence of side effects [24]. 

In recent studies, the gut microbiota has been identified as one of the environmental factors that may intervene in the etiopathogenesis of MS [25] as it is involved in the modulation of the host immune system, alters the permeability of the BBB, promotes autoimmune demyelination, and interacts directly with different cell types in the central nervous system. This occurs through interaction with gut cells, production of metabolites by microbes, secretion of gut hormones, and changes in neural and immune signaling and blood circulation to the BBB. Evidence of these interactions has been found in animal models but also in human cell cultures in vitro [26,27]. 

Interestingly, gut bacteria may also influence the integrity of the BBB [28]. This evidence suggests that MS patients may more frequently present gut microbial dysbiosis than healthy controls and that the gut microbiota may be potentially operative in predisposing or modifying the disease course of MS [25]. 

### 1.2. Microbiota and Multiple Sclerosis

Several studies using *16S rRNA* gene sequencing to perform a detailed analysis of the fecal microbiome have revealed that MS patients have a different microbial community profile than healthy controls, supporting the hypothesis that the gut microbiota may be an environmental etiopathogenetic factor in MS. Indeed, recent studies have shown a decrease in certain populations of beneficial bacteria that make up the gut microbiota in MS patients compared to healthy subjects, confirming that gut dysbiosis is a constant in the clinical course of MS [8]. 

The gut microbiota is the set of micro-organisms (archaea, fungi, bacteria, and eukaryotes) that populates the gastrointestinal tract [29] and includes the collective genome of approximately 100 trillion resident microorganisms [30]. It plays a very important role in the human organism, being responsible for immunoregulation, host metabolism, digestion, vitamin synthesis, neurodevelopment, and energy homeostasis [31,32,33,34,35,36]. 

The composition of the gut microbiota varies throughout a person’s lifetime, being influenced by various factors such as diet and the energy sources they are provided with [37], medication (particularly antibiotics), and lifestyle; indeed, the fetal gastrointestinal tract (GIT) is sterile before birth and is first colonized by microbes during birth [38]. 

In the general population, the gut microbiota is composed of different microorganisms, including bacteria, yeasts, and viruses. The two phyla *Firmicutes* and *Bacteroidetes* alone account for about 90% of the gut microbiota, followed by *Actinobacteria*, *Proteobacteria*, *Fusobacteria*, and *Verrucomicrobia*. The phylum *Firmicutes* is composed of more than 200 different genera, such as *Lactobacillus*, *Bacillus*, *Clostridium*, *Enterococcus*, and *Ruminicoccus*, and the phylum *Bacteroidetes* mainly of *Bacteroides* and *Prevotella* [39]. 

Several scientific studies have investigated the composition of the gut microbiota in MS patients [40,41]. Cosorich et al. compared the microbiota of patients with remitting relapsing MS (MS-RR) with that of healthy subjects, observing an increase in the phylum Firmicutes and a reduction in the phylum *Bacteroides* in MS patients in the relapsing phase, compared to healthy subjects and MS patients in the remission phase [42]. In addition, an increase in *Streptococcus mitis* and *Streptococcus oralis* and a decrease in *Prevotella* species, which normally produces the anti-inflammatory metabolite propionate, has been observed [43,44,45]. A reduction of it, on the other hand, correlates with the expansion of Th17 cells and disease activity [8]. *Streptococcus mitis* is also able to induce the differentiation of Th17 cells and is involved in immune-mediated tissue damage [46]. Furthermore, a decrease in *Clostridium* genera was observed in MS-RR patients compared to healthy subjects; this decrease leads to a lower production of short-chain fatty acids and a reduction in regulatory T cells (Treg), resulting in a lower production of the anti-inflammatory cytokine IL-10 [44,47,48]. It has been shown that the microbiota is directly involved in the pathogenesis of MS by regulating the proliferation of Th17 cells in the intestine [49]. 

Th17 cells, together with regulatory T cells (Tregs), are in dynamic balance and cooperate in maintaining the homeostasis of the immune microenvironment and gut health. Specifically, Th17 cells, under the condition of homeostasis, secrete small amounts of IL-17 and IL-22 by promoting epithelial cell proliferation and upregulating the expression of antimicrobial peptides and tight junction proteins, thus protecting the intestinal mucosa from external insults [50]. When the diversity and composition of the gut microbiota are disrupted because of environmental factors or the regulation of susceptibility genes, pathogenic bacteria can promote colonization and infection by breaking down mucin-like nutrients, releasing toxic gases, and secreting enzymes to destroy mucosal barriers. In these cases, there is an increase in the number of Th17 cells with IL-17 expression in the foci of inflammation [50]. 

In addition, IL17 is an interleukin that can compromise the BBB through, at least in part, reactive oxygen formation. Furthermore, human MS patient endothelial cells have been shown to express high levels of receptors for IL-17, perhaps facilitating transmigration into the CNS independent of BBB disruption. When Th17 cells are increased and IL-17 is overproduced, dysregulation of brain tight junctions occurs, and infiltration of Th17 cells into the CNS is facilitated by dense expression of CCR6 on the cell surface. CCR6 interacts with CCL20, which is highly expressed at sites of inflammation. IL-17 further amplifies this process by inducing the expression of CCL20. Neuronal damage is promoted, directly, because of apoptotic granzyme B released by myelin-specific Th17 cells and indirectly, as IL-17 stimulates CNS resident cells to produce inflammatory mediators [51]. 

In conclusion, a higher frequency of Th17 cells and mRNA levels of IL-17 were found in brain lesions of MS patients than in healthy controls. In addition, correlations were found between disease activity and the amounts of Th17 and IL-17 in MS lesions. Specifically, more than 70% of IL-17-producing T cells were observed in active MS lesions compared with 17% in inactive plaques [51]. 

In a study performed by Jangi et al., an increase in *Methanobrevibacter* (phylum *Euryarchaeota*) and *Akkermansia* (phylum *Verrucomicrobia*), and a decrease in *Butyricimonas*, were observed in MS patients [52]. *Methanobrevibacters* are distributed at the lymphoid areas of the intestinal mucosa [53] and appear to be associated with various inflammatory bowel diseases. Furthermore, these bacteria may have a proinflammatory action related to their ability to degrade mucus, causing damage to the intestinal barrier, and increasing the exposure of intestinal immune cells to microbial antigens [54]. The *Butyricimonas* genus produces butyrate, a short-chain fatty acid that induces the proliferation of Treg cells. A reduction in butyrate production is observed in numerous autoimmune diseases [55], as this condition alters intestinal barrier function and promotes inflammation. 

Several studies report an increase in *Proteobacteria* in MS, which also occurs in other autoimmune diseases such as Inflammatory Bowel Disease (IBD). They may contribute to the onset of autoimmune diseases by promoting the inflammatory response [56]. All these studies confirm that dysbiosis in the clinical course of MS is regularly present [57]. 

As mentioned before, diet is one of the main factors regulating the composition and function of the gut microbiome, since acute changes in diet can alter the composition of the gut microbiome within 24 h, through alteration of metabolic pathways and microbic gene expression [58,59]. Scientific evidence suggests that high-fat diets, refined food, and low-fiber diets modify the microbiome by promoting the onset of an inflammatory state. On the contrary, nutrients such as omega-3, polyunsaturated fatty acids, fibers, vitamin D3, and polyphenols can modify the microbiota with beneficial effects on the body, favoring the proliferation of bacterial microorganisms producing anti-inflammatory substances [8]. Finally, a further link between the microbiome and MS derives from studies correlating the risk of MS with antibiotic use. A lower risk of MS has been observed for penicillin users, but data are uncertain and need to be further confirmed [60].

### 1.3. Polyphenols

Polyphenols constitute a heterogeneous group of natural substances, particularly known for their positive effect on human health, sometimes referred to as vitamin P.

In nature, polyphenols are produced by the secondary metabolism of plants, where in relation to their chemical diversity, they play different roles: defense against herbivorous animals (imparting an unpleasant taste) and pathogens (phytoalexins), mechanical support (lignin) and as a barrier against microbial invasion, attraction for pollinators and fruit dispersal (anthocyanins).

Chemically, polyphenols are molecules composed of several condensed phenolic cycles (organic compounds possessing one or more hydroxyl groups—OH—linked to an aromatic ring). According to their structure, they can be schematically divided into different classes: phenolic acids, flavonoids, stilbenes, and lignans (Figure 1) [61,62]. 

Phenolic acids have at least a hydroxyl group (-OH) on a benzene ring slightly ionized and are soluble in water, corrosive, and irritating to the skin. They are reactive compounds that are easily oxidized, forming complexes with metals or polymers. Phenolic acids are widely distributed in foods and beverages [63]. 

Flavonoids are compounds that are widely distributed in plants and consist of two aromatic rings linked by a three-carbon atom chain. They are derived from phenylalanine and malonyl-CoA (through the fatty acid pathway). They have many -OH groups on ring structures which give them antioxidant properties as H- or electron donors, and they can also act as metal chelators [64]. Polymers of flavonoids, known as condensed tannins, are present in plants and help to give the plant structure and defense against pathogen attacks. Structural variations in the rings allow flavonoids to be subdivided into different families: chalcones, flavanones, flavones, flavanols, anthocyanidins, and isoflavones. Quercetin (onion, apple, grape, broccoli, tea), luteolin (lemon, olives, celery), and epigallocatechin (tea) are very present in the diet in significant amounts [65]. 

Stilbenes are phenolic compounds that contain two benzene rings separated by an ethane or ethene bridge derived from the shikimate and malonate pathways. In the plant, they function as antimicrobials and growth regulators. The major source of stilbenes in the diet is resveratrol, found in black grapes and red wine. Its concentration in red wine as well as that of polyphenols in general may depend on several factors, the main one being the winemaking technique (the maceration-fermentation stage) [66,67]. 

Lignans, formed by two 2-phenyl propane units, are of particular interest due to their pharmacological and anti-cancer, anti-viral properties against venereal diseases such as those caused by the *Papilloma virus*. The largest source of lignans in the diet is linseed (>3.7 g/kg dry weight) [68]. 

A further group of polyphenols have a chemical structure consisting of two aryl groups joined by a chain of seven carbon atoms; they can be cyclic or linear, e.g., myricanone and curcumin, respectively [69].

A large number of studies have shown a correlation between polyphenol intake and a decrease in risk factors for chronic diseases. Polyphenols as flavonoids and stilbenes interact with ATP-binding cassette molecules, reducing drug resistance in cancer [70,71]. However, their low bioavailability, due to interaction with the food matrix, liver-mediated metabolic processes (phase I and II metabolism), the gut, and the microbiota, limits their positive effects [72]. After being ingested, polyphenols are recognized as xenobiotics [73] and act as natural antioxidants due to their metal-chelating and free radical scavenger properties [64]. Furthermore, polyphenols can influence the cells of the immune system and modulate the expression of cytokines and pro-inflammatory genes [74]. Their properties depend on bioactive metabolites, produced when they are metabolized by the microbiota [75]. In vitro and animal model studies have recently shown that consumption of polyphenol-rich foods positively affects intestinal permeability by strengthening the barrier properties of the intestinal epithelium and increasing the synthesis and expression of tight junctions. In addition, resveratrol influences the intestinal transport of glucose, alanine, and chloride by PKA/cAMP pathway activation [76]. Despite this, the mechanisms by which polyphenols act remain unclear [77,78]. 

### 1.4. Microbiota and Polyphenols

Diet has a leading role in modulating the intestinal microbiota [79]. Dietary polyphenols have prebiotic-like effects by selectively stimulating beneficial bacteria and reducing the incidence of diseases [80]. Specifically, they can stimulate some keystone beneficial bacteria species, such as *Akkermansia muciniphila*, *Bacteroides thetaiotaomicrom*, *Faecalibacterium prausnitzii*, *Bifidobacteria*, and *Lactobacilli* [81,82,83], while reducing the number of pathogens, such as *C. Perfringens* and *C. Histolyticum* [84]. 

Due to the reciprocal interaction between polyphenols and microbiota, it is possible not only to maintain gut health by modulating the growth of beneficial bacteria and inhibiting the growth of pathogens [85], but also to increase the bioavailability of the first one. 

Only 5–10% of dietary polyphenols consumed are absorbed, while 90–95% (typically flavonoid aglycones and polymers) reach the colon, where they interact with commensal microbiota and show their antimicrobial and prebiotic effects [86,87]. Thus, there are auxiliary health benefits of dietary polyphenols which may be attributed to their metabolites produced by gut microbiota.

All those dietary factors, or metabolites of them, that can promote immune cell differentiation and the production of regulatory, rather than inflammatory, cytokines thus have the potential ability to reduce the formation of new inflammatory lesions and clinical relapse using pathways like traditional MS disease-modifying therapies. In addition, foods that can attenuate neuroinflammation and that fight oxidative stress or protect mitochondria may help prevent chronic demyelination and axonal/neuronal damage [18]. 

Although it is not our intention to propose substituting food components for specific drugs, we believe that some of them can be of great help in enhancing the beneficial effects of drugs. 

This review summarizes current knowledge on the molecular mechanisms by which specific polyphenols exert neuroprotective effects, with reference to the pathogenetic mechanisms of multiple sclerosis. The influence of dietary polyphenols in the relapse and progression of multiple sclerosis will be explored as well. 

## 2. Molecular Mechanisms of Polyphenol Action on Neuroinflammation and Demyelination Pathways

Nutritional status can influence the course of MS. It is known that ROS have physiological effects on numerous signal pathways, but an excess of them can lead to oxidative stress [88]. Dietary nutrients can exacerbate or reduce symptoms by modulating inflammatory and redox-related pathways [89]. In addition, the redox state is strictly linked to energy metabolism which is a source of ROS [90,91]. 

Being overweight/obesity is associated with a low-level chronic inflammatory state which could be a contributing risk factor in some neuroinflammatory diseases like MS [92]. Dietary nutrients can interact with multiple enzymes, transcription factors, and receptors, inducing physiological and metabolic (catabolic or anabolic) cellular responses. Activation of these processes modulates inflammation and autoimmune responses [93]. 

On the other hand, anabolic diets, rich in fat and simple carbohydrates, combined with a lack of exercise, induce pro-inflammatory pathways downstream of the nuclear transcription factor-kB (NF-kB) through the activation of the sterol regulatory element-binding protein 1c and 2 (SREBP) and the carbohydrate reactive element binding protein (ChREBP). On the other hand, exercise associated with catabolic diets, characterized by calorie restriction and the intake of antioxidant molecules, such as polyphenols found in fruit and vegetables and long-chain omega-3 polyunsaturated fats (n-3) acids (PUFA) found in fish, induce the anti-inflammatory AMPK/Sirtuins/PPAR pathway, counteracting the activation of the transcription factor NF-kB [13]. 

Different biomolecular events modulated by diet contribute to the onset and progression of multiple sclerosis, and among them, oxidative stress plays a key role [94,95]. 

In many diseases, including viral [96,97], cerebrovascular [98], neurodegenerative, and autoimmune diseases like MS [99], the primary generator of ROS leading to oxidative stress is membrane NADPH-oxidase (NOX). 

NADPH oxidase is also involved in neuron and glial cell function [99]. Oligodendrocyte progenitor cells (OPCs), which are present in the adult CNS, proliferate, migrate, and differentiate into myelinating oligodendrocytes (OLs) [100] to ensure a continuous turnover of mature OLs and remyelination that occurs at the demyelinating plaques in MS patients. Within chronic demyelinating lesions of MS patients, there is a small number of mature OLs and an increased number of OPCs suggesting that in a pathological condition, the OPCs maturation process is slowed [101,102]. Oxidative stress and the presence of pro-inflammatory molecules at the level of focal areas of demyelination can explain, at least in part, the impairment of oligodendrocyte differentiation and remyelination failure. 

Despite the proven efficacy of DMT, polyphenols could represent synergistic therapeutic agents for the treatment of MS for their antioxidant and anti-inflammatory effects, hardly associated with DMT.

Some studies highlighted the molecular mechanisms involved in the antioxidant and anti-inflammatory effects of foods rich in polyphenols relevant for the reversion of pathological processes leading to demyelination and neurodegeneration in multiple sclerosis. Significant data from studies in vitro and in animal models of MS have been collected for resveratrol, curcumin, luteolin, quercetin, and hydroxytyrosol, and therefore we will focus on the results obtained with these polyphenols (Figure 2). 

### 2.1. Resveratrol

Resveratrol (3, 40, 5-trihy- droxystilbene) is a polyphenolic compound and a member of the non-flavonoid subclass, mainly found in red grapes, blueberries, rhubarb, mulberries, pistachios, and peanuts [103,104,105]. It is a phytoalexin produced by plants during environmental stress and pathogen-induced mechanical damage, and this explains why its main properties were initially thought to be fungitoxic and protective against disease in plants [106]. In recent years, it has been shown that resveratrol possesses many other biological properties, in particular, it may have antioxidant, anti-inflammatory, anti-apoptotic, anticarcinogenic, anti-ageing, and osteogenic effects [107,108]. Due to these properties, resveratrol has important neuroprotective and cardioprotective effects [109,110]. However, its main limitations appear to be its poor bioavailability, rapid absorption, and low solubility in water which reduce its use in vivo [111]. For this reason, new strategies have been implemented to increase its bioavailability, such as the use of nanostructured lipid carriers (NLCs) and solid lipid nanoparticles (SLNs) [112,113]. 

Due to its strong antioxidant and anti-inflammatory activities, resveratrol appears to be an important candidate for the treatment of inflammatory neurodegenerative diseases such as MS. Several data have been obtained using the experimental autoimmune encephalomyelitis (EAE) animal model of MS, in which the main characteristic is neuroinflammation arising after immunization with myelin-derived antigens. This model presents notable limitations [114,115]. However, it is noteworthy that EAE studies with germ-free animals have been useful in clarifying potential mechanisms by which gut microbiota may influence the course of the disease [116]. 

One of the major molecular targets of resveratrol is sirtuin 1 (SIRT1), an NAD+-dependent deacetylase that regulates inflammatory responses and promotes mitochondrial function by modulating essential metabolic regulatory transcription factors [117]. Nimmagadda et al. have demonstrated that the overexpression of SIRT1 in EAE transgenic mice has an immunomodulatory and neuroprotective effect by reducing the levels of the proinflammatory cytokines IL-17 and IFN-γ, and by increasing the levels of IL-10. The activation of caspase 3, involved in the apoptotic pathway, is also reduced. Analyses of myelin basic protein (MBP) levels and the number of neurofilaments (NF) also showed that, in this animal model, demyelination and axonal damage occur to a lesser extent [118]. The pharmaceutical-grade formulation of resveratrol SRT501, with enhanced systemic absorption, activates SIRT1, preventing neuronal damage and long-term neurological dysfunction in EAE mice. The protective effects of SRT501 can be ascribed at least in part to SIRT1 activation, since the administration of sirtinol, a SIRT1 inhibitor, attenuated SRT501 neuroprotective effects in EAE mice [119]. Another study confirmed that SRT501 delays the onset of EAE in mice and prevents neuronal damage. Moreover, SRT501 administration suppressed inflammation in chronic but not in relapsing EAE [120]. Further effects of SRT501 in EAE mice are the reduction of oxidative stress by suppressing the overexpression of NADPH oxidases, NOX2, and NOX4, and its ability to maintain the integrity of the BBB by protecting the tight junction proteins of the basement membrane [121].

Recently, an in vivo study performed in EAE mice showed that silicate resveratrol derivatives, designed to increase its bioavailability, reduced the incidence, severity, and progression of the disease with increased efficacy compared with resveratrol [122]. Experiments performed using the cuprizone-induced demyelination animal model of MS confirmed the protective effects of resveratrol in MS. Unlike EAE which mimics mainly the immune-related events in MS, the cuprizone model reflects the de/re-myelination processes occurring in MS and therefore is a useful tool to evaluate the effects of substances on remyelination and repair mechanisms in MS [123]. The administration of resveratrol in cuprizone-treated C57Bl/6 mice significantly improved their clinical condition by reducing oxidative stress, mitochondrial dysfunction, and activation of NF-kB signaling. Moreover, resveratrol exhibited neuroprotective effects, increasing the differentiation of OPCs and remyelination inside cuprizone-induced white matter lesions [94].

In a recent paper, Gandy et al. highlighted that resveratrol epigenetically ameliorates EAE development through increasing miR-124 levels. Higher levels of miR-124 consequently inhibit sphingosine kinase 1 (SK1) expression specifically in the brain, thereby improving the disease condition [124]. Importantly, this brain-specific miRNA is decreased in MS mice models and patients [125].

### 2.2. Curcumin

Curcuma Longa or turmeric is a plant belonging to the family of *Zingiberaceae*, known as the “golden spice” or “spice of life”, with powerful antioxidant, anti-inflammatory, anticancer, and antimicrobial effects. For this reason, it is widely used not only in cooking but also in medicine for the treatment of gastrointestinal and hepatic disorders, cardiovascular and many neurodegenerative diseases such as MS. Curcumin, the active component of turmeric, is one of the most studied polyphenols because it prevents protein aggregation, maintains the homeostasis of the inflammatory system, and removes free radicals and toxic aggregates from the brain [126].

Several studies conducted both in vitro and in vivo, as well as clinical studies, have demonstrated the safety of curcumin for the treatment of many diseases. Thanks to its properties, curcumin can have a protective effect on MS, since it can act on different possible targets thus improving the clinical manifestations of this disease [127]. Curcumin improves neurological symptomatology and the infiltration of immune cells in the spinal cord of EAE mice and ameliorates the behavioral defects and demyelination in the corpus callosum of cuprizone-treated mice. In vitro, curcumin reduces the apoptosis of oligodendrocytes and exerts an inhibitory effect on CD4+ T-cell proliferation and differentiation, reducing the levels of secreted proinflammatory cytokines, including IL-6, IL-1β, TGF-β, TNF-α, and others. Curcumin increased serum levels of anti-inflammatory molecules such as TGF-β and IL-10 in EAE mice treated with curcumin [128]. In addition, curcumin may inhibit the inflammatory response mediated by microglia by inactivating the AXL/JAK2/STAT3 signaling pathway [129]. Another study evidenced that the administration of polymerized nano-curcumin with improved solubility diminishes EAE symptoms through downregulation of the expression of pro-inflammatory and oxidative stress markers and enhancing neurotrophic factors and myelin repair [130]. 

In addition, Lu et al. evidenced that, in EAE mice, curcumin-loaded high-density lipoprotein-mimicking peptide-phospholipid scaffold (HPPS) (Cur-HPPS) is captured by monocytes through the scavenger receptor class B type I (SR-B1), hampering monocytes in crossing the BBB and blocking the proliferation of microglia cells. This immunomodulatory effect responsible for a significative reduction in morbidity in EAE models is due to the inactivation of NF-kB and its downstream pathway, and to the downregulation of the expression of adhesion-and migration-related molecules [131]. 

In EAE, an augmented expression of some receptors of Toll-like family (TLR) has been observed. These receptors play a key role in innate immunity by increasing pro-inflammatory molecules and are also essential modulators of the autoimmune process [132]. In EAE mice, curcumin modulates the expression of TLR. The administration of curcumin led to a reduction in the activation of TLR4 and TLR9 receptors in CD4+ and CD8+ T-cells [133,134]. Finally, it has been demonstrated that in the brains of cuprizone mice, curcumin treatment not only increased the activity of antioxidant enzymes, such as superoxide dismutase (SOD) and catalase (CAT), and the levels of glutathione (GSH) but decreased demyelination [128].

### 2.3. Luteolin

Luteolin (3’,4’,5,7-tetrahydroxy-flavone) is a compound belonging to the flavonoid subclass and is considered one of the most bioactive polyphenols [135]. This glycosylated flavonoid is present in fruit and vegetables, particularly oranges, carrots, celery, broccoli, parsley, and chamomile [136]. It has powerful anti-inflammatory, antioxidant, antitumour, and antimicrobial effects associated with ROS and RNS scavenging activity [137]. It has been shown that the use of luteolin combined with the composite ultramicronised palmitoylethanolamide, an endogenous N-acylethanolamine, significantly reduces the development of clinical signs in EAE mice by modulating the transcript expression of proteins associated with the inflammatory process such as serum amyloid A1 (SAA1), TNF-α, IL-1β, and IFN-γ [138]. 

Blocking the release of inflammatory mediators in diseases such as Alzheimer’s Disease (AD), Parkinson’s Disease (PD), Amyotrophic Lateral Sclerosis (ALS), and MS, luteolin exerts a high neuroprotective effect [139]. Specifically, it can inhibit the release of IL-6, IL-8, and TNF-α [140] and suppress the NF-kB, STAT3, c-Jun N-terminal kinase (JNK), and p38 signaling pathways and all ERK1/2-regulated kinases involved in glial cell activation and release of inflammatory mediators from mast cells [139,141]. Mast cells are concentrated near blood vessels in the brain and regulate the permeability of the BBB during the immune response and neuroinflammation [142]. 

Proteases produced by mast cells are able to cause direct damage to myelin; in fact, a higher level of this enzyme has been found in the cerebrospinal fluid (CSF) of MS patients [143]. In vivo studies have shown that luteolin is able to block the onset of experimental allergic EAE [144] and to prevent the release of IL-1, TNF-α, and metalloproteinase-9 (MMP-9) from peripheral blood mononuclear cells (PBMCs) of MS patients [145]. It has been seen that the release of MBP or other myelin degradation products induces mast cell degranulation. Kempuraj et al. showed that MBP is able to promote the release of inflammation mediators such as IL-6, IL-8, TGF-β1, TNF-α, vascular endothelial growth factor (VEGF), histamine, and tryptase by human mast cells and that luteolin is able to inhibit the release of these pro-inflammatory molecules. The inhibitory abilities of luteolin on the processes described suggest its possible use to reduce the inflammatory state that characterizes MS [141]. The main limitation to the use of this polyphenol remains its low bioavailability and its low solubility in water: one approach to solving this problem could be the production of polymerized luteolin nanoparticles [146]. 

### 2.4. Quercetin

Quercetin (3,3′4′,5,7-pentahydroxy flavone), or quercitin, is a flavonol belonging to the flavonoid subclass naturally occurring in a wide variety of fruits (apples, grapes, olives, citrus fruits, berries), vegetables (tomatoes, onions, broccoli, capers), and medicinal herbs (lovage leaves, dill, coriander, elderberry, and ginkgo biloba) [147,148]. Quercetin is insoluble or poorly soluble in water, depending on the temperature, but very soluble in lipids and alcohol. The bioavailability of quercetin is very low due to its absorption, varying between 3% and 17%, extensive metabolism, and/or rapid elimination [149]. Like the other polyphenols, quercetin also exhibits antioxidant activity by scavenging free radicals, chelating metal ions, and inhibiting lipid peroxidation. Quercetin also shows anti-inflammatory, antimicrobial, especially limiting the growth of pathogenic bacteria, and antitumor activity [148]. 

In vitro treatment of mononuclear cells from the blood of MS patients with this polyphenol showed a reduction in IL-1β, MMP-9, and TNF-α production and NF-kB inhibition [150]. In addition, Hendriks et al., have demonstrated that quercetin can reduce ROS levels and prevent phagocytosis of myelin by macrophages [151]. In EAE models, quercetin treatment inhibited IL-12 production by Th1 cells and blocks NF-kB, Janus kinase 2 (JAK2), tyrosine kinase 2 (TYK2), STAT3, and STAT4 signaling mostly by inhibiting tyrosine kinase activities [152]. These promising results suggest a potential use of quercetin for the treatment of MS. 

### 2.5. Hydroxytyrosol

It is known that the consumption of olive oil, the main source of monounsaturated fats in the MD, is beneficial for the prevention of several diseases. Hydroxytyrosol (HT), together with tyrosol, oleuropein, oleocanthal, and oleacein, are the main polyphenols in olive oil [153]. This amphipathic phenol, mainly present in the leaves of olive trees (Olea europea L.) and olives, where it determines the bitter taste, is rapidly metabolized and absorbed [154] and, therefore, its bioavailability is extremely low [155]. 

HT is one of the most powerful antioxidants found in nature and has also been shown to have anti-inflammatory, anti-carcinogenic, anti-atherogenic, and anti-ageing effects [156]. The main activity of this polyphenol is the intracellular and extracellular scavenging of ROS.

Due to these important properties, the neuroprotective and cardioprotective effects of HT have been widely investigated. The ability to cross the BBB [157,158] is certainly a key feature in explaining the beneficial effects of this polyphenol in the CNS. Several studies conducted on animal models of neurodegenerative diseases such as MS, where redox imbalance and inflammation play a key role not only in the brain but also at the systemic level, evidenced the neuroprotective potential of HT. Indeed, it has been demonstrated that the administration of HT in EAE rat models results in an antioxidant effect in the spinal cord and peripheral blood, reducing lipid peroxidation products and enhancing the role of glutathione peroxidase (GPx). In these animals, HT also reduces oxidative stress in the gastrointestinal system and other organs by limiting toxin production by the gut microbiota [159]. In addition, HT reduces the expression levels and activity of MMP-9 and MMP-2 enzymes in activated rat astrocytes and in the serum of MS patients [160,161]. This effect is relevant for MS since immune cells, such as Th17 cells and macrophages, can cross the BBB and activate the inflammatory cascade in the CNS, causing myelin degradation and neuronal damage in MS [162]. The increased permeability of the BBB is due to an increased expression of MMP-9, which facilitates leukocyte infiltration into the CNS. Nutritional intervention with this polyphenol could therefore become of considerable relevance in the treatment of MS.

**Figure 2 ijms-24-07247-f002:**
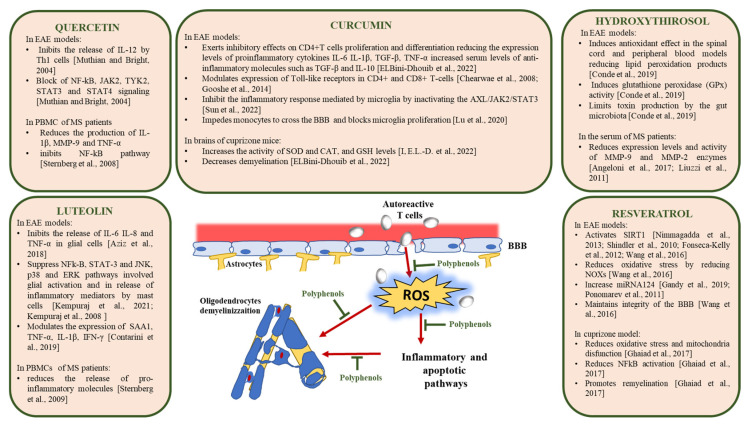
Molecular mechanisms of action of polyphenols in MS [94,118,119,120,121,124,125,128,129,131,133,134,139,140,141,145,150,152,159,160,161].

## 3. Polyphenols in Multiple Sclerosis: Clinical Evidence

Despite the fairly large number of studies conducted in vitro and with animal models of MS, clinical evidence on the protective effects of polyphenols in MS patients is still limited to a few compounds. The results of clinical trials aimed at studying the effects of curcumin and epigallocatechin gallate in MS subjects will be reviewed below. 

### 3.1. Curcumin

Curcumin can cross the blood-brain barrier, so it exerts its neuroprotective role through direct effects. Various clinical studies demonstrate the therapeutic effects of curcumin in MS patients. Nanocurcumin may inhibit disease progression in MS patients, showing neuroprotective effects by regulation of inflammatory gene expression. Dolati et al. [163] investigated the effects of a six-month oral administration of nanocurcumin in patients with relapsing-remitting multiple sclerosis (RRMS) on the expression levels of inflammation-related molecules including microRNAs (miRNAs), miRNA-dependent targets, transcription factors, and pro-inflammatory cytokines in PBMCs. The results showed that nanocurcumin decreases the expression levels of IL-6, IL-1β, INF-γ, activator protein 1 (AP-1), STAT1, C-C motif chemokine ligand 2 and 5 (CCL2, CCL5), TNF-α, and NF-κB. Moreover, nanocurcumin treatment reduced the expression of miRNA-145, miRNA-132, and miRNA-16. Consequently, nanocurcumin increases the expression of miRNA-dependent targets including Sox2, a factor promoting remyelination, sirtuin-1 which is implicated in inflammatory responses, forkhead box P3 (Foxp3), and programmed cell death 1 (PDCD1) which controls the immune system. In addition, nanocurcumin decreased IFN–γ, CCL2, and CCL5 secretion levels. Generally, the expanded disability status scale (EDSS) score was significantly improved in the group supplemented with nanocurcumin compared with the placebo group. Another clinical study has demonstrated that 6 months of nanocurcumin therapy in patients with RRMS is able to restore the expression profile of specific miRNAs, evaluated by quantitative PCR in PBMCs [164]. Altered expression of several miRNAs, mostly those influencing immune homeostasis, has been considered responsible for the pathogenesis of MS [165]. However, the direct implication of these miRNA changes upon nanocurcumin treatments has not been fully elucidated yet. 

In addition, the effects nanocurcumin on the frequency of Th17 lymphocytes and on the expression of transcription factor and associated cytokines have also been analyzed by flowcytometry, real-time PCR, and ELISA, respectively. After 6 months of oral administration of nanocurcumin, MS patients showed a significant decrease in Th17-associated parameters, such as Th17 frequency, RAR-related orphan receptor γt (RORγt), and IL-17 expression and secretion levels [166]. 

The role of T-regulatory (Treg) cells in the pathogenesis of CNS autoimmune inflammatory diseases, including MS, is known. Another study showed that nanocurcumin regulates immune system function and prevents autoreactivity through an increase in the frequency and function of Treg cells. Furthermore, nanocurcumin treatment significantly increases the Treg-associated cytokine levels, including TGF-β and IL-10. In line with previous reports, the EDSS score and the quality of life of patients were significantly improved [167]. 

### 3.2. Epigallocatechin Gallate (EGCG)

EGCG, the main polyphenol found in green tea, has antioxidant and anti-inflammatory properties [168] and enhanced energy metabolism and substrate utilization [169]. Since MS is characterized by fatigue and muscle weakness, Mähler et al., in a double-blind placebo-controlled crossover trial, examined metabolic response to EGCG treatment for 12 weeks, with a 4-week washout period in between in RRMS male and female patients [170]. Fasting and postprandial energy expenditure (EE), as well as fat oxidation (FAOx) and carbohydrate oxidation rates (CHOx), were measured at rest or during 40 min of exercise (0.5 W/kg), using indirect calorimetry. At rest, blood and microdialysis samples were also extracted from adipose tissue and skeletal muscle to evaluate local tissue perfusion and metabolism. Results evidenced different responses to EGCG treatment in men and women, probably due to the sex-specific influence of autonomic and endocrine systems. At rest, the postprandial EE, CHOx, glucose intake, and adipose tissue perfusion were significantly lower in men, but higher in women treated with EGCG compared to placebo. During exercise, postprandial EE was lower in EGCG-treated patients compared to placebo. After EGCG treatment, an increase in CHOx during exercise was observed in men, but not in women. These data indicate that EGCG administered to MS patients enhances muscle metabolism during moderate exercise to a greater extent in men than in women. 

In addition, EGCG seems to have the potential to improve cardiac risk in MS patients. Benlloch et al. analyzed the impact of intervention with EGCG and ketone bodies on cardiovascular risk in MS patients by examining anthropometric variables and specific biomarkers of risk. MS patients were randomly assigned to either a control group or an intervention group. Subjects of both groups were given an isocaloric diet for 4 months. In the intervention group, the diet was enriched with 60 mL of extra virgin coconut oil and supplemented with 800 mg of EGCG while the control group received the placebo. After 4 months of treatment, there was a significant increase in levels of paraoxonase 1, an enzyme that inhibits low-density lipoprotein oxidation, and albumin in the intervention group; there was also a significant decrease in waist-to-hip ratio, as well as a significant increase in muscle mass. These changes were not observed in the control group. The improvement in both anthropometric measurements as well as in serum analytes determined a decrease in cardiovascular risk [171]. 

## 4. Conclusions

MS is a multifactorial complex disease where various environmental factors including nutrition and metabolism alteration have been recognized as important elements in the onset and progression of the disease. Among dietary interventions experimented with in MS patients, the Mediterranean diet seems to be the most feasible and free from side effects [24]. There are currently still too few and sometimes inconsistent studies regarding the effects of diet on the treatment of MS [172], and therefore there is a need to potentiate research in this direction.

Although there is broad agreement on the importance of dietary supplements like polyphenols with antioxidant, anti-inflammatory, and neuroprotective effects as adjuvant therapy in MS, often this therapeutic tool is not considered as due, in part because the molecular mechanisms underlying the effects are not always well understood. The anti-inflammatory and antioxidant effects of polyphenols have been more extensively characterized while the neuroprotective effects have been well documented for resveratrol and quercetin, whereby research must be extended to the other compounds belonging to this large group of phytochemicals. Another limit to the use of polyphenols in the treatment strategy of MS is their poor bioavailability, even if currently a great effort is being made in formulating preparations with greater bioavailability. Besides the use of lipid carriers, further possible strategies, already experimented with for resveratrol and curcumin to increase their solubility, stability, and bioactivity, could be encapsulation in protein nanocomplexes, insertion in polymeric nanoparticles, solid dispersions, and nanocrystals [112,173]. 

Even if, according to evidence-based studies, at present no dietary intervention can replace DMT in becoming an alternative approach to treatment, the literature data collected in this review related to the molecular mechanisms of polyphenols in vitro and in animal models of MS, like EAE and cuprizone treated mice, strongly encourage the use of dietary polyphenols as a novel therapeutic approach for MS. Nevertheless, clinical trials evaluating the effects of polyphenol administration in MS patients are scarce, conducted on a small number of patients, and restricted only to a few molecules. 

We believe that an even better knowledge of the molecular mechanisms by which dietary polyphenols affect the inflammatory status and neuroprotection in MS, and more clinical data are needed to provide greater awareness of the correct use of these substances as nutritional interventions in MS. 

## Figures and Tables

**Figure 1 ijms-24-07247-f001:**
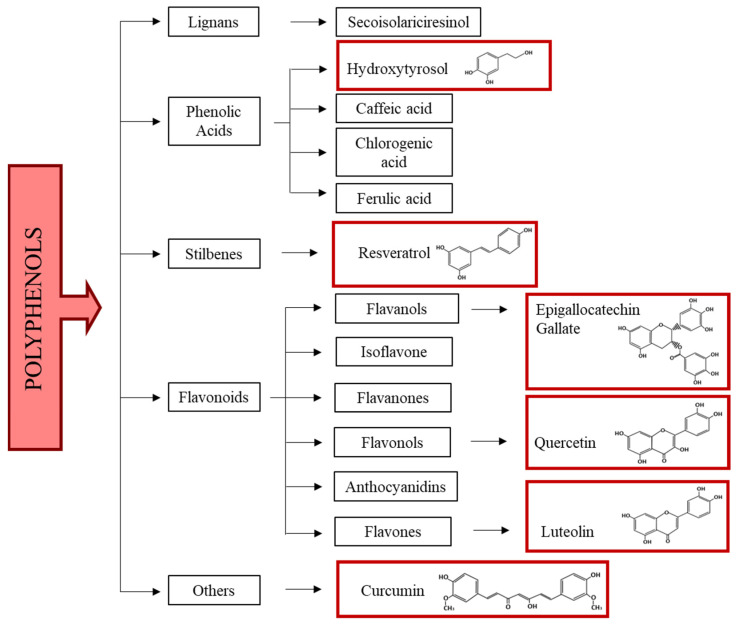
Polyphenol classification. In red boxes the chemical formula of polyphenols used in in vitro and in vivo models of MS and in clinical trials are reported.

## Data Availability

Not applicable.

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
