# Peer review of "Dietary Polyphenols, Microbiome, and Multiple Sclerosis: From Molecular Anti-Inflammatory and Neuroprotective Mechanisms to Clinical Evidence"

_ijms, 2023, doi:10.3390/ijms24087247_

Round 1

Reviewer 1 Report

Dear authors

yor review on the use of polyphenols as adjunctive treatment of MS is valid. Tha references and mechanisms of action of the various polyphenols are adequately chosen. I therefore think teh revew can be published. Maybe, in Discussion mention natural but well prescribed Mediteranian diet as an option for clinical trials with MS and other similar patients. Also, you can discuss how to increase the bioavailability of these substances apart from liposomes and other lipid nanoparticles.

Author Response

We thank the reviewer for his/her helpful suggestions. In the manuscript changes are marked in red. We have reported literature data about the use of Mediterranean diet in MS patients in the text (paragraph 1.1, lines 127-137) and we have included a comment on this issue in the Conclusions (lines 634-637) In the conclusions we have also mentioned other possible methods to improve the bioavailability of polyphenols in general (lines 648-651).

Reviewer 2 Report

In their proposed review, La Rosa and her team are offering some unconventional (or, perhaps, “non-mainstream”) answers to the riddle of multiple sclerosis. The ideas are not new, however, the combination of selected naturally occurring compounds (polyphenols) and intestinal microflora does produce a potentially attractive topic. In principle, the review is well organized, and the language is fluent; at times, however, the text is too wordy (e.g. # 5 but also elsewhere) and unnecessarily speculative with apparently unreferenced and/or unsupported statements (as in, but not limited to, # 20). Despite good English, there are occasional slips and errors in spelling and grammar (typos?) that should have been picked and fixed by an automatic check or by careful reading (examples ## 2 & 21).

Later in my report, I am including a sample of recent review articles which could make it easier to update and streamline the reference list (see ## 6 & 17 and at the end). Keeping the citations and the list of references properly targeted and rigorously up to date seems imperative in a good review, never mind whether it conforms or not to the prevailing ideology.

Specific queries:

#   1     Lines 1 and 2: The review covers intestinal microbiome almost as much as the dietary polyphenols; this should be reflected in the title. I am aware of the line 23 in Abstract.

#   2     Line 20: “increasing”

#   3     Line 40: Where? In the world, in the US, EU.. ? Per year, ten years? Prevalence or incidence? Source? What is the ratio of female/male case of MS? I am told that there are other mysterious factors, such as the latitude where one grew up (?). Could this impact the microbiome?

#  4   Line 41: It would help to individually reference at least some of the following statements in the paragraph.

#   5   Line 55: Many of the neurological diseases which we do not understand and do not know how to cure or treat are called "multifactorial". I would start simply by "In MS, inflammation of the CNS is the primary cause of.. “

#   6     Line 66: This is a place where authors might consider updating the reference list.

#   7     Line 70: Is there any evidence that the gut bacteria have a similar effect on BBB in humans?

# 10     Line 84: 100 trillion species or organisms?

# 11   Line 104: How direct? How specific is the role of Th17 cells in MS? More recent evidence/confirmation?

# 12    Line 165: “black grapes and red wine” – has this been thoroughly investigated? Which variety of grapes, which red wine (Merlot, Cabernet Sauvignon, appellation d’origine)? Is the effect dose-dependent? References?

# 13      Line 177: Are they handled by ATP binding cassette molecules?

# 14      Line 182: The effect on the morphology of the intestinal wall is interesting enough. However, resveratrol has also been shown to influence glutamate transport in brain (starting at micromolar concentrations; Vieira de Almeida et al Cell Mol Neurobiol 27 661-668 (2007) doi: 10.1007/s10571-007-9162-2) – have the authors searched for possible effects of resveratrol on the intestinal transporters of amino acids (and other biological molecules)?

# 15      Line 218: Reference?

# 16      Lines 226/227: Reference? Does not excess adiposity (being overweight) entail, in itself, a low-level chronic inflammation which could be a contributory risk factor in some neurological etiopathologies? MS?

# 17   Line 288 (and the following lines): Using EAE as an animal model of MS has attracted criticism. Authors might consider citing a recent review by Melamed at al. doi 10.3389/fnmol.2022.1019877

# 18      Line 343: Which type of SOD?

# 19      Lines 409-411: Reference?

# 20      Lines 444/445: Reference? Overall, some of the statements in this and the following paragraph are not only unreferenced (except for [125]) but appear too uncertain/speculative. Are the changes in the miRNA's a part of the etiology of MS or just an epiphenomenon (consequence of the disease)? Incidentally, one could question the whole conjecture of an involvement of microbiome in MS along the same lines (cause or effect?). I wonder whether the authors can comment.

# 21      Line 446: “ ..study has..”

# 22      It occurs to me that, if the microbiome is important in the etiopathogenesis of MS, there should be a relationship between taking antibiotics and the risk of MS. The only reference I was able to find was Alonso A (2006) doi: 10.1093/aje/kwj123 which seems to imply that the damage to gut microbiome protects against MS, perhaps. Is there any more information on the subject? This could be crucial.

Two other recent reviews to consider:

Bonnechere et al (2022) IJMS 23 13665 doi: 103390/ijms2322113665

Dziedzic & Saluk (2022) IJMS 23 14478 doi: 103390/ijms2322114478

Author Response

We wish to thank the reviewer for his/her constructive remarks regarding our manuscript. We have revised the manuscript according to recommendations. In the manuscript changes are marked in red.

Specific queries:

#   1     Lines 1 and 2: The review covers intestinal microbiome almost as much as the dietary polyphenols; this should be reflected in the title. I am aware of the line 23 in Abstract.

We thank the reviewer for appropriately suggesting to include the word “Microbiome” in the title   given the space dedicated in the review to this aspect of the pathogenesis of multiple sclerosis and its interrelationships with diet and nutraceuticals

#   2     Line 20: “increasing”

it has been corrected

#   3     Line 40: Where? In the world, in the US, EU.. ? Per year, ten years? Prevalence or incidence? Source? What is the ratio of female/male case of MS? I am told that there are other mysterious factors, such as the latitude where one grew up (?). Could this impact the microbiome?

As suggested, we have included more epidemiological data of  MS and positive association between latitude and prevalence in the text (paragraph 1.1 lines 39-50). We included three new references: Walton, C. et al 2020;  Steve Simpson Jr et al 2011 and Heather Wood 2016

#  4      Line 41: It would help to individually reference at least some of the following statements in the paragraph.

The sentences have been individually referenced.

#   5     Line 55: Many of the neurological diseases which we do not understand and do not know how to cure or treat are called "multifactorial". I would start simply by "In MS, inflammation of the CNS is the primary cause of.. “

We modified the text as suggested (paragraph 1.1 line 112).

#   6     Line 66: This is a place where authors might consider updating the reference list.

The references have been updated with more recent ones (paragraph 1.1 line 139).

#   7     Line 70: Is there any evidence that the gut bacteria have a similar effect on BBB in humans?

New references related to the link between microbiota and BBB in mice has been added (paragraph 1.1 lines 142-145) Ordoñez-Rodriguez, A., et al.2023 and Tang, W. et al 2020. To the best of our knowledge data in human is lacking.

# 10     Line 84: 100 trillion species or organisms?

We changed “species” with “microrganisms” (paragraph 1.2 line 160)

# 11     Line 104: How direct? How specific is the role of Th17 cells in MS? More recent evidence/confirmation?

An in-depth discussion of the role of Th17 in MS can be found in paragraph 1.2 (lines 189-215).We added two new references: Chen, L. et al 2023 and   Moser, T. et al. 2020

# 12     Line 165: “black grapes and red wine” – has this been thoroughly investigated? Which variety of grapes, which red wine (Merlot, Cabernet Sauvignon, appellation d’origine)? Is the effect dose-dependent? References?

“The concentration of resveratrol in red wine as well as that of polyphenols in general may depend on several factors, the main one being the winemaking technique” We added in the text this sentence with the relative references: Eva López-Rituerto et al 2021 and  Wang S. et al 2014 (paragraph 1.3 lines 272-274). We did not include data about the red wines or black grape varieties with the higher concentration of resveratrol because they are uncertain.

# 13     Line 177: Are they handled by ATP binding cassette molecules?

Some polyphenols interact with ATP binding cassette molecules. We added this evidence and the relative references in the text (paragraph 1.3 line 283-284). Michalak and  Wesolowska, 2012 and Bachmeier, et al 2009.

# 14     Line 182: The effect on the morphology of the intestinal wall is interesting enough. However, resveratrol has also been shown to influence glutamate transport in brain (starting at micromolar concentrations; Vieira de Almeida et al Cell Mol Neurobiol 27 661-668 (2007) doi: 10.1007/s10571-007-9162-2) – have the authors searched for possible effects of resveratrol on the intestinal transporters of amino acids (and other biological molecules)?

As suggested, we discussed the evidence related to the influence of resveratrol on intestinal transports. We added the  reference:  Klinger  and Breves 2018 (paragraph 1.3 line 295-296).

# 15     Line 218: Reference?

We have added a new reference published in 2023 in IJMS that deals with the link between metabolism and ROS generating systems (Pecchillo et al 2023).

# 16     Lines 226/227: Reference? Does not excess adiposity (being overweight) entail, in itself, a low-level chronic inflammation which could be a contributory risk factor in some neurological etiopathologies? MS?

We have added the concept that overweight/obesity is associated with a low-grade inflammatory state which represents an additional risk factor for neurodegenerative disease. (paragraph 2 Lines 338-339). We added the refereces: Correale, and Marrodan 2022, 8paragraph 2 Lines 339) and Inés Bravo-Ruiz et al 2021 (paragraph 2 Lines 342)

# 17     Line 288 (and the following lines): Using EAE as an animal model of MS has attracted criticism. Authors might consider citing a recent review by Melamed at al. doi 10.3389/fnmol.2022.1019877

We have cited the very interesting review by Melamed et al in the text (paragraph 2.1 lines 401),  reporting the limitations of the EAE model and also how the germ-free model of EAE has allowed to highlight the role of microbiome in the onset and in the modulation of severity degree of the disease. We added the references Ransohoff, R. M. 2012 and Rudick  et al 2013. (paragraph 2.1 lines 396-401).

# 18     Line 343: Which type of SOD?

The authors of the paper cited in our review have measured tissue total SOD enzymatic activity, and presumably it refers to all SOD isoforms. Accordingly, we have not modified the text.

# 19     Lines 409-411: Reference?

The references De Leonardis 2007, Gallardo-Fernández et al 2019, and Carloni et al. 2018,  have been added in paraghraph 2.5 line 536-543.

# 20     Lines 444/445: Reference? Overall, some of the statements in this and the following paragraph are not only unreferenced (except for [125]) but appear too uncertain/speculative. Are the changes in the miRNA's a part of the etiology of MS or just an epiphenomenon (consequence of the disease)? Incidentally, one could question the whole conjecture of an involvement of microbiome in MS along the same lines (cause or effect?). I wonder whether the authors can comment.

We thank the Reviewer for the helpful comment. We include in the text the importance of microRNAs in MS etiology and add a reference focusing on microRNA in immune homeostasis of MS (paragraph 3.1 lines 579-585). Wang and Liang 2022.

# 21     Line 446: “ ..study has..”

It has been corrected

# 22     It occurs to me that, if the microbiome is important in the etiopathogenesis of MS, there should be a relationship between taking antibiotics and the risk of MS. The only reference I was able to find was Alonso A (2006) doi: 10.1093/aje/kwj123 which seems to imply that the damage to gut microbiome protects against MS, perhaps. Is there any more information on the subject? This could be crucial.

We also reported in the paper data about the use of antibiotics and risk MS citing Alonso et al., 2006, as suggested (paragraph 1.2 lines 242).

Two other recent reviews to consider:

Bonnechere et al (2022) IJMS 23 13665 doi: 103390/ijms2322113665

Dziedzic & Saluk (2022) IJMS 23 14478 doi: 103390/ijms2322114478

Both the review papers have been cited (paragraph 1.2 Lines 174).

Reviewer 3 Report

The article is a narrative review on the role of dietary polyphenols in multiple sclerosis (MS). Despite recent progress in therapeutic options in MS, further investigation is being conducted in this field, in search of enhancing efficacy and safety of current approved therapies, as well as new targets for treatment. Dietary compounds are one of the subjects of these investigations, especially natural substances available in food products, which are quite popular among people with MS. However, clinical evidence of their beneficial effects is still scarce, but certainly deserves attention.

The topic of the review is  therefore interesting and up-to-date. It contains (especially in the chapters 2 and 3) a comprehensive description of polyphenols (resveratrol, curcumin, luteolin, quercetin and hydroxytyrosol), their properties and possible role in MS, well illustrated with Fig. 1 and 2. The findings from experimental models and clinical studies (integrating immunological/metabolomic/epigenetic markers with  clinical outcomes) are clearly and consistently presented . Limitations of polyphenols potential use as therapeutic agents and future direction of studies were appropriately addressed. Large number of references, including most recent ones, was used and correctly cited.

However, the first part of the review (chapter 1) has some serious shortcomings, which impede its quality and didactic value. The following issues need to be completed, clarified or revised:

1.      The most common clinical symptoms of the disease should be described, highlighting their diversity and emerging disability (The brief sentence : “Clinical manifestations include depression, cognitive impairment, and intestinal dysfunction” is much oversimplified and not consistent with characteristic of MS)

2.      Beside thoroughly described inflammatory demyelination, neurodegeneration with axonal loss should be presented as one of the main component of MS background. The role of oxidative stress and apoptosis should be stressed, considering  that “molecular mechanisms of neuroprotection” is the main topic of the study and was indeed thoroughly discussed for polyphenols in chapters 2 and 3

3.      Except for the brief statement in the abstract, the introduction is completely lacking any data about current state of MS treatment. First of all, information should be provided on available therapies in MS  (disease modifying therapies – DMT) and recent progress in this field. Next, some challenges and unmet needs of these therapies (e.g. little evidence for neuroprotective effects and/or stopping progressive phase of disease) could be mentioned, which encourage further investigation of new therapeutic agents. See e.g.:

Wiendl H, Gold R, Berger T, et al. Multiple Sclerosis Therapy Consensus Group (MSTCG): position statement on disease-modifying therapies for multiple sclerosis (white paper). Therapeutic Advances in Neurological Disorders. 2021;14. doi:10.1177/17562864211039648

Amin M, Hersh CM. Updates and advances in multiple sclerosis neurotherapeutics. Neurodegener Dis Manag. 2023 Feb;13(1):47-70. doi: 10.2217/nmt-2021-0058. Epub 2022 Oct 31. PMID: 36314777.

Finally, it has to be stressed that, according to evidence-based studies,  no dietary intervention can replace DMT or become an alternative approach to treatment. There is no evidence for “nutritional intervention alleviating possible side effects of immune-modulatory drugs  and the symptoms of MS patients”, either. Thus  in conclusions these issues could be only outlined as the future directions of investigation

4.      The impact of diet upon risk and course of MS is not limited to interactions with gut microbiom. See e.g.  Fitzgerald K. J Neurol Neurosurg Psychiatry 2018;89:3  ,

Katz Sand I. The Role of Diet in Multiple Sclerosis: Mechanistic Connections and Current Evidence. Curr Nutr Rep. 2018 Sep;7(3):150-160. doi: 10.1007/s13668-018-0236-z. PMID: 30117071; PMCID: PMC6132382)

Other putative mechanisms should be reviewed, to be more consistent with description of polyphenols mode of action in chapter 3. Despite focus on gut microbiom in chapter 1, there is only a brief summary of links between its composition and polyphenols (paragraph 1.4) , and this thread is not continued at all in the chapters 2 and 3.

5.      Aim of study is not sufficiently justified or supported in the  introduction (again some relevant points only appear in the abstract). What is the reason for studies on dietary compounds in MS? Why have polyphenols been chosen as the subject for this review? Why did the aim of review  focus on neuroprotective mechanisms ?(while, otherwise, immunomodulatory effects are convincingly presented as well)

6.      Consequently, the possible effects of polyphenols in MS should be outlined as: neuroprotective, anti-inflammatory/immunoregulatory and promoting demyelination. It would be convenient to point out, which ones predominate for particular polyphenols, and summarize them in the conclusions. 

Author Response

We thank the reviewer for his/her valuable comments. We highly appreciated his/her opinion and suggestions. In the manuscript changes are marked in red.

  1. The most common clinical symptoms of the disease should be described, highlighting their diversity and emerging disability (The brief sentence: “Clinical manifestations include depression, cognitive impairment, and intestinal dysfunction” is much oversimplified and not consistent with characteristic of MS)

Clinical symptoms of MS have been better described as suggested (paragraph 1.1 lines 50-61). We added the references: Wiendl et al 2021 ,Noyes & Weinstock-Guttman, B. 2013 and  Tafti, at al. 2022

  1. Beside thoroughly described inflammatory demyelination, neurodegeneration with axonal loss should be presented as one of the main component of MS background. The role of oxidative stress and apoptosis should be stressed, considering that “molecular mechanisms of neuroprotection” is the main topic of the study and was indeed thoroughly discussed for polyphenols in chapters 2 and 3

Following the suggestions, neurodegeneration accompanying inflammatory demyelination has been better illustrated (paragraph 1.1 Line 72-77). We added four references:  Adamczyk, B.; Adamczyk-Sowa 2016, Friese, et al 2014, Beckman and   Koppenol 1996, and Gonsette, 2008

  1. Except for the brief statement in the abstract, the introduction is completely lacking any data about current state of MS treatment. First of all, information should be provided on available therapies in MS (disease modifying therapies – DMT) and recent progress in this field. Next, some challenges and unmet needs of these therapies (e.g. little evidence for neuroprotective effects and/or stopping progressive phase of disease) could be mentioned, which encourage further investigation of new therapeutic agents. See e.g.: Wiendl H, Gold R, Berger T, et al. Multiple Sclerosis Therapy Consensus Group (MSTCG): position statement on disease-modifying therapies for multiple sclerosis (white paper). Therapeutic Advances in Neurological Disorders. 2021;14. doi:10.1177/17562864211039648; Amin M, Hersh CM. Updates and advances in multiple sclerosis neurotherapeutics. Neurodegener Dis Manag. 2023 Feb;13(1):47-70. doi: 10.2217/nmt-2021-0058. Epub 2022 Oct 31. PMID: 36314777.

Although an extensive description of the existing therapies for MS is beyond the scope of the review, we agree with the reviewer that an overview of drugs currently used is needed (see paragraph 1.1 lines 81-111). We added the suggested references and more references: Hauser & Cree 2020, Katz Sand I. 2018, and Gholamzad et al 2019.

Finally, it has to be stressed that, according to evidence-based studies, no dietary intervention can replace DMT or become an alternative approach to treatment. There is no evidence for “nutritional intervention alleviating possible side effects of immune-modulatory drugs and the symptoms of MS patients”, either. Thus in conclusions these issues could be only outlined as the future directions of investigation

The sentence “alleviating possible side effects of immune-modulatory drugs  and the symptoms of MS patients” in the context of discussion was, in the authors' mind, speculative. However, it has been cancelled. Although we believe that the notion that at present nutritional intervention with polyphenols  cannot replace current DMT has been expressed in multiple points of the  review,  to emphasise the concept we modified discussion introducing the sentence suggested by the reviewer paragraph 4 (lines 653-654)

  1. The impact of diet upon risk and course of MS is not limited to interactions with gut microbiom. See e.g. Fitzgerald K. J Neurol Neurosurg Psychiatry 2018;89:3  ,Katz Sand I. The Role of Diet in Multiple Sclerosis: Mechanistic Connections and Current Evidence. Curr Nutr Rep. 2018 Sep;7(3):150-160. doi: 10.1007/s13668-018-0236-z. PMID: 30117071; PMCID: PMC6132382)

We added the two suggesterd references and three more references to comment this issue (paragraph 1.1 lines 126-137). Atabilen and AkdevelioÄŸlu,  2022, Jayasinghe  et al 2022, and Noormohammadi, et al 2022.

Other putative mechanisms should be reviewed, to be more consistent with description of polyphenols mode of action in chapter 3. Despite focus on gut microbiom in chapter 1, there is only a brief summary of links between its composition and polyphenols (paragraph 1.4) , and this thread is not continued at all in the chapters 2 and 3.

In various points of the review, more emphasis is placed on the role of the microbiome and its interaction with polyphenols. Accordingly, the word “microbiome” has been added in the title.

  1. Aim of study is not sufficiently justified or supported in the introduction (again some relevant points only appear in the abstract). What is the reason for studies on dietary compounds in MS? Why have polyphenols been chosen as the subject for this review? Why did the aim of review  focus on neuroprotective mechanisms ?(while, otherwise, immunomodulatory effects are convincingly presented as well)

We better described in the introduction the reason for reviewing polyphenols as side therapy for multiple sclerosis (paragraph 2 lines 366-369).

  1. Consequently, the possible effects of polyphenols in MS should be outlined as: neuroprotective, anti-inflammatory/immunoregulatory and promoting demyelination. It would be convenient to point out, which ones predominate for particular polyphenols, and summarize them in the conclusions.

We have outlined that antioxidant and anti-inflammatory effects are documented for all polyphenolic compounds, while neuroprotective effects have been demonstrated only for few substances, as resveratrol or quercetin. (paragraph 4 lines 642-646).

Round 2

Reviewer 2 Report

Authors made a number of useful modifications as suggested in my report. I checked all of the changes. Some new passages in the new version sound a bit awkward; English needs to be corrected (lines 241/242; line 43 “In order” is out of place here). Please, check again the grammar and spelling throughout. Apart from that I am satisfied with the changes resulting from my comments.

Author Response

Authors made a number of useful modifications as suggested in my report. I checked all of the changes. Some new passages in the new version sound a bit awkward; English needs to be corrected (lines 241/242; line 43 “In order” is out of place here). Please, check again the grammar and spelling throughout. Apart from that I am satisfied with the changes resulting from my comments.

We modified the text according to the suggestions and performed a grammar and spelling check

Reviewer 3 Report

The majority of my remarks have been satisfactorily addressed and consequently these aspects of the manuscript were indeed improved. I have only two minor concerns:

1)      Although the overview of DMT is sufficient and relevant, statements about their “major side effects” and apparently low safety  seem exaggerated. E.g:

Page 3: “Side effects associated with currently used therapies are divided into 'desirable', which require monitoring as they involve immunomodulation/immunosuppression, and  'undesirable', which cause hepatotoxicity, cardiotoxicity, development of Progressive Multifocal Leukoencephalopathy (PML) and allergic reactions [4]. For these reasons, the development of treatment protocols with substances with a  higher health safety profile is mandatory”.

Page 8: “Since there is no etiological therapy for MS and DMT are often accompanied by major  side effects, an increasing number of researches are focused on the study of the effects of  natural substances as polyphenols with powerful antioxidant, anti-inflammatory and  neuroprotective effects as side therapy for MS”.

Observational studies provide evidence for good safety profile for the majority of high-efficacy therapies and the recommendations for monitoring and prevention allow to further reduce the risks. Furthermore, there is no evidence that polyphenols (or other natural substances) could ameliorate or prevent the side effects of DMT.  So, instead of these,   potential antioxidant and neuroprotective effects of polyphenols (hardly addressed by DMT) should be more highlighted as the reason for their expected beneficial  use as synergistic therapeutic agents

2)      Links between diet, gut microbiom and MS background have been extensively described. However, adding “microbiom “ in the title is not necessary. I would suggest rather:

Dietary Polyphenols and Multiple Sclerosis: Molecular anti-inflammatory and neuroprotective  mechanisms and Clinical Evidence

Author Response

Although the overview of DMT is sufficient and relevant, statements about their “major side effects” and apparently low safety  seem exaggerated. E.g: Page 3: “Side effects associated with currently used therapies are divided into 'desirable', which require monitoring as they involve immunomodulation/immunosuppression, and  'undesirable', which cause hepatotoxicity, cardiotoxicity, development of Progressive Multifocal Leukoencephalopathy (PML) and allergic reactions [4]. For these reasons, the development of treatment protocols with substances with a  higher health safety profile is mandatory”.

We have eliminated Progressive Multifocal Leukoencephalopathy (PML) among “undesirable side effects” since it is a rare complication encountered in patients treated with a select number of disease-modifying therapies (DMTs). Moreover, the sentence “For these reasons, the development of treatment protocols with substances with a  higher health safety profile is mandatory” has been eliminated.

Page 8: “Since there is no etiological therapy for MS and DMT are often accompanied by major  side effects, an increasing number of researches are focused on the study of the effects of  natural substances as polyphenols with powerful antioxidant, anti-inflammatory and  neuroprotective effects as side therapy for MS”. Observational studies provide evidence for good safety profile for the majority of high-efficacy therapies and the recommendations for monitoring and prevention allow to further reduce the risks. Furthermore, there is no evidence that polyphenols (or other natural substances) could ameliorate or prevent the side effects of DMT.  So, instead of these,   potential antioxidant and neuroprotective effects of polyphenols (hardly addressed by DMT) should be more highlighted as the reason for their expected beneficial  use as synergistic therapeutic agents

According to the reviewer suggestion, we change the sentence as follow:

“Despite the proven efficacy of DMT, polyphenols could represent synergistic therapeutic agents for the treatment of MS for their antioxidant and antinflammatory effects, hardly associated with DMT”.

2)      Links between diet, gut microbiom and MS background have been extensively described. However, adding “microbiom “ in the title is not necessary. I would suggest rather: Dietary Polyphenols and Multiple Sclerosis: Molecular anti-inflammatory and neuroprotective  mechanisms and Clinical Evidence

We have changed the title as suggested by the reviewer. However we kept “microbiome” in the title as suggested by another referee because in the new version of the manuscript the role of microbiome in multiple sclerosis pathogenesis has been greatly expanded.